# Computer-Vision-Based Sensing Technologies for Livestock Body Dimension Measurement: A Survey

**DOI:** 10.3390/s24051504

**Published:** 2024-02-26

**Authors:** Weihong Ma, Yi Sun, Xiangyu Qi, Xianglong Xue, Kaixuan Chang, Zhankang Xu, Mingyu Li, Rong Wang, Rui Meng, Qifeng Li

**Affiliations:** 1Information Technology Research Center, Beijing Academy of Agriculture and Forestry Sciences, Beijing 100097, China; mawh@nercita.org.cn (W.M.); xuexl@nercita.org.cn (X.X.); a91616@163.com (K.C.); limy@nercita.org.cn (M.L.); mengr@nercita.org.cn (R.M.); 2College of Information Engineering, Northwest A&F University, Xianyang 712199, China; sunyi@nwafu.edu.cn (Y.S.); 1043283468@nwafu.edu.cn (Z.X.); rongw@nwafu.edu.cn (R.W.); 3College of Information and Electrical Engineering, China Agricultural University, Beijing 100083, China; qxy@cau.edu.cn

**Keywords:** computer vision sensing, live body dimension measurement, 3D point cloud, image processing

## Abstract

Livestock’s live body dimensions are a pivotal indicator of economic output. Manual measurement is labor-intensive and time-consuming, often eliciting stress responses in the livestock. With the advancement of computer technology, the techniques for livestock live body dimension measurement have progressed rapidly, yielding significant research achievements. This paper presents a comprehensive review of the recent advancements in livestock live body dimension measurement, emphasizing the crucial role of computer-vision-based sensors. The discussion covers three main aspects: sensing data acquisition, sensing data processing, and sensing data analysis. The common techniques and measurement procedures in, and the current research status of, live body dimension measurement are introduced, along with a comparative analysis of their respective merits and drawbacks. Livestock data acquisition is the initial phase of live body dimension measurement, where sensors are employed as data collection equipment to obtain information conducive to precise measurements. Subsequently, the acquired data undergo processing, leveraging techniques such as 3D vision technology, computer graphics, image processing, and deep learning to calculate the measurements accurately. Lastly, this paper addresses the existing challenges within the domain of livestock live body dimension measurement in the livestock industry, highlighting the potential contributions of computer-vision-based sensors. Moreover, it predicts the potential development trends in the realm of high-throughput live body dimension measurement techniques for livestock.

## 1. Introduction

In the domain of animal husbandry, it is commonly believed that the size and physique of livestock can reflect the superior characteristics of their breeds [1]. In the process of enhancing livestock breeding techniques and selecting superior livestock breeds to increase production, automated morphometric measurement technology is indispensable [2]. However, the livestock industry has long been plagued by issues such as inadequate technological contributions, low resource utilization efficiency, inconsistent product quality, insufficient value conversion, and challenges in realizing environmental benefits, all of which have severely impeded the process of agricultural modernization [3].

Presently, the livestock industry is undergoing rapid development. Overseeing livestock physiology encompasses various aspects, including temperature monitoring, body weight measurement, and live body dimension assessment. In recent years, with the advancement of scientific technology, live body dimension measurement in the realm of animal husbandry has garnered considerable attention. The dimensions of livestock hold substantial importance as indicators for breeding facilities. Nevertheless, at present, the majority of farms continue to rely on manual methods involving tape measures, measuring rods, and other tools for the manual measurement of cattle, sheep, and other livestock. This approach is not only limited by errors stemming from tools and human subjectivity but also frequently triggers stress responses in animals, resulting in significant stress consequences. This is particularly pronounced in the context of large-scale breeding facilities, where the efficiency of manual measurement is notably insufficient both in terms of time and space considerations.

In recent years, propelled by the rapid advancement of scientific and technological progress, the integration of information technology with various industries has become increasingly profound. With the incorporation of computer vision sensing, this integration has reached new heights. Various advanced computer technologies such as big data, image processing, and computer vision have gradually permeated across diverse sectors [4]. These developments have given rise to a plethora of products that not only enhance productivity in societal and economic activities but also elevate the overall quality of life [5]. The livestock industry has consistently remained a crucial sector for agricultural efficiency enhancement and rural income augmentation [6]. In the realm of information technology being applied to animal husbandry, the technology for measuring the live body dimensions of livestock typically utilizes sensors to acquire livestock data using methods such as scanning and imaging. Subsequently, computer processing is applied to yield the live body dimension measurements of the animals [7]. This approach is usually non-invasive, mitigating stress reactions in livestock and thus constituting a non-destructive testing method [8].

To date, researchers have made significant advancements in the measurement of live body dimensions in various domesticated animals such as pigs, cattle, and sheep utilizing computer vision sensing technology. Commendable results have been achieved in this regard [9]. This paper conducts a comprehensive review and summary of the relevant research in this domain. The survey encompasses three key aspects: livestock data collection (sensing data acquisition), livestock data processing (sensing data processing), and data analysis of live body dimension parameters (sensing data analysis), as illustrated in Figure 1. According to the above three aspects, we have analyzed the current challenges faced in automatic livestock body measurement, respectively, based on criteria such as the timeliness of the measurement process, the accuracy of the measurement results, and the robustness of the measurement technology. This paper also anticipates the main development trends in this field for the future.

## 2. Raw Sensing Data Acquisition for Live Livestock Body Dimensions

The primary step in the live body dimension measurement of livestock is the collection of livestock data. Currently, livestock data acquisition commonly employs sensing devices such as depth cameras, 3D scanners, or 2D RGB cameras, as depicted in Table 1. Subsequently, suitable equipment is utilized to capture images of cattle, sheep, and other livestock in standing or walking states, thereby acquiring corresponding livestock data. Livestock data acquisition primarily considers two aspects: the acquisition equipment and the acquisition method, both of which exert a certain influence on the precision of the subsequent live body dimension measurements.

### 2.1. Sensing Technology for Collecting Animal Body Measurements

The selection of sensing livestock acquisition equipment is typically influenced by factors such as price, acquisition environment, and the subjects being captured. Yang et al. (2022) [10] used a Huawei P20 smartphone to capture images of cows and employed Structure from Motion (SfM) technology to transform the farm environment and cow images into 3D point clouds. While the camera cost was low, the 3D reconstruction process proved time-consuming. Li et al. (2022) [11] used five Kinect DK cameras to capture and reconstruct images of beef cattle, achieving swift 3D model reconstruction but with sensitivity to lighting conditions. Zhao et al. (2015) [12] employed Kinect depth cameras to acquire color and depth images of sheep bodies, considering lighting influences and necessitating precise alignment between the measured objects and sensing sources.

As the livestock industry evolves, live body dimension measurement technology is progressively moving toward industrialization and commercialization. In the current phase, the preferred characteristics for equipment selection in this field tend to prioritize stability, high accuracy, and low cost.

Currently, most of the livestock acquisition equipment employs depth cameras, a pioneering approach that maximizes animal welfare and provides improved means to assess animal health and reactions [13,14]. Depth cameras, also referred to as 3D cameras, discern the depth distances within the captured space, setting them apart significantly from standard RGB cameras. The presently used depth cameras can be categorized into three types based on their operating principles: binocular stereo vision, structured-light, and time-of-flight (TOF), as illustrated in Figure 2. Table 2 contrasts the advantages and disadvantages of these camera types, along with their key manufacturers, based on their fundamental principles. Table 3 further compares these cameras based on aspects like resolution, accuracy, frame rate, lighting conditions, and cost. Binocular stereo vision, despite its resilience to lighting, often encounters image discrepancies due to changing lighting conditions, leading to matching failures or reduced accuracy in practice [15]. TOF cameras exhibit lower noise at longer distances and boast higher frame rates, rendering them more suitable for dynamic scenes [16]. Structured-light technology is power-efficient, mature, and better suited to static scenes [17]. Live body dimension intelligent measurement in the livestock domain primarily occurs in dynamic environments, making TOF cameras a more suitable choice due to their capacity to meet the data acquisition requirements, especially for such scenarios.

### 2.2. The Methods of Livestock Live Body Dimension Measurement

Currently, livestock data collection methods can primarily be categorized into three types: channel archway style, suspended fixed style, and simple portable style. Table 4 provides a comparison of the advantages and disadvantages of these three collection methods. When collecting data within livestock farms, it is essential to consider the specific environmental context and opt for an appropriate collection approach.

The channel archway method for livestock data collection typically involves constructing an archway-style collection device next to the pathway where the livestock walk and then capturing data as the livestock pass through the pathway, as depicted in Figure 3a. Ruchay et al. (2020) [18] employed three Kinect v2 cameras installed in the left, right, and top positions of the pathway. However, due to the narrow railing of the pathway, certain safety concerns were present. Li et al. (2022) [19] established an archway device with five Kinect DK cameras positioned in the top, upper-left, lower-left, upper-right, and lower-right positions. The railing in the middle of the archway was thicker to address equipment and personnel safety issues, but the data collection was somewhat compromised due to the railing obstruction.

During the collection process, when cows passed through the collection area, the equipment first recognized the ear tag numbers of the cows and then initiated data collection using the five depth cameras while the cows were in relatively stationary conditions. This collection method not only saves time and effort but also ensures rapid data acquisition. However, if the archway is sufficiently large and multiple cattle pass through simultaneously, occlusion issues may arise. Furthermore, given the diverse species and varied poses of livestock, there is a higher demand for algorithmic sophistication.

Using the suspended fixed-style method for livestock data collection typically involves installing a data collection device directly above the livestock and guiding the livestock to stand beneath the device for data capture, as shown in Figure 3b. Ye et al. (2022) [20] employed depth cameras to capture top-down depth video data on beef cattle, followed by frame-by-frame processing of the video data. Kamchen et al. (2021) [21] utilized depth cameras to capture images of the backs of livestock and subsequently employed a multilayer perceptron neural network to select the highest-quality cattle image data. This collection method generally requires only one depth camera, significantly reducing costs. However, since it only captures data from the back of the cattle, the measurement coverage is limited and might not fulfill the measurement requirements for the other body parts of the livestock.

Using the simple portable-style method for livestock data collection typically involves utilizing readily available cameras or simple devices, as depicted in Figure 3c. This approach is suitable for complex farm environments. Shi et al. (2022) [22] used drones to capture images and performed aerial triangulation calculations based on the images’ inherent position and orientation system (POS) information. After obtaining the initial coordinate point cloud data through adjustment, three-dimensional reconstruction was conducted. Although this process takes longer, it boasts low camera costs and rapid initialization. Pezzuolo et al. (2018) [23] created simple devices by placing Kinect v1 cameras above and next to the pig feeding area, capturing imaging data from both the side and the top of the animals. The simple portable-style method is often limited to capturing data from individual livestock and might encounter challenges such as data loss and low efficiency. It may struggle to meet the demands of large-scale livestock farms.

In summary, the process of livestock body dimension measurement is significantly reliant on livestock data collection. Hence, during the livestock data collection phase, it is imperative to consider factors such as the rearing environment and the subject of measurement in order to select suitable collection methods and sensing devices. This approach ensures the acquisition of high-precision, information-rich, and low-noise raw livestock data.

## 3. Processing of Raw Sensing Data

During the collection of point cloud data from livestock, we commonly utilize depth cameras, regular RGB cameras, or other sensing devices to obtain three-dimensional point cloud data or image data on the livestock. As a result, the data processing for livestock mainly involves image data and point cloud data.

### 3.1. Processing of Livestock Image Sensing Data

Currently, digital image processing mainly involves methods such as denoising, image equalization, image filtering, and edge contour detection. These techniques play a crucial role in the correction, enhancement, and analysis of livestock image data.

#### 3.1.1. Image Detection

Livestock image detection refers to the technology that employs computer processing and analysis of livestock images in various complex environments, utilizing deep learning algorithms to identify different livestock objects.

Brahim et al. (2020) [24] captured images of cow heads in a cowshed and used a convolutional neural network (CNN)-based model for individual identification. This CNN-based approach yielded excellent results even in complex farm environments, achieving an accuracy of 97% in identification. However, this method solely relying on the head for recognition cannot be applied to images containing only the body of a cow. Hu et al. (2020) [25] obtained a set of side-view images of cows and performed cow localization in each image. They then divided the images into three parts: head, torso, and legs. Deep features were extracted from each part using a CNN, and a feature fusion strategy was designed to combine these features. This approach achieved a recognition accuracy of 98.36% for cows, as illustrated in Figure 4. This method not only identifies cows based on their heads but also utilizes other parts of their bodies, leading to an improved recognition accuracy. Shen et al. (2020) [26] proposed an automated algorithm for cow recognition using side-view images. Based on the AlexNet model, the experimental results demonstrated the superiority of this method over traditional recognition methods and deep learning methods focusing on local regions. This method exhibits promising application prospects, offering a robust approach to image-based livestock recognition.

Livestock image detection can accurately and efficiently identify specific livestock in complex environments. This capability allows for the selection of appropriate measurement methods for different livestock species in subsequent body measurement processes, ultimately enhancing the accuracy of livestock body measurements.

#### 3.1.2. Image Segmentation

Image segmentation for livestock refers to the process of separating livestock targets from the background in an image. Initially, computers identify livestock objects in the image and then extract various specific image features to differentiate the livestock from the background.

Rotimi-Williams et al. (2021) [27] proposed an enhanced Mask R-CNN instance segmentation method to support the extraction of cattle with blurred boundaries and irregular shapes. This approach mitigates the negative effects of lighting variations during cattle image capture, reducing the misclassification of pixels differentiating between actual cattle bodies and shadows. Liu et al. (2020) [28] introduced the use of Gaussian Mixture Models to separate cows from the background, as depicted in Figure 5. Additionally, applying Gaussian Mixture Models to depth images was found to address challenges in target detection due to background variations, offering a new approach to livestock object extraction based on images.

Image segmentation for livestock can effectively identify diverse livestock objects in various complex environments, laying an essential foundation for subsequent image-based measurements of livestock dimensions.

#### 3.1.3. Image Posture Judgment

In various environments and situations, the same livestock can exhibit different poses. To assess the various poses of the same livestock in images, computer technologies such as deep learning are often employed. These technologies help recognize the diverse poses of livestock.

Martin et al. (2020) [29] designed a deep learning system for position and pose detection, using the Faster R-CNN object detection pipeline and Neural Architecture Search (NAS) as the underlying network for feature extraction. This system can differentiate between standing and lying pigs, as shown in Figure 6, where the blue bounding boxes indicate standing pigs and the green ones indicate lying pigs. The method achieves an average precision of 80.2% for position and pose detection with a sufficient number of training images, but the precision is lower for datasets with limited training images. Jessy et al. (2022) [30] extended the open-source pose estimation toolkit DeepLabCut, demonstrating its powerful capabilities in animal pose estimation, identification, and tracking using datasets of varying complexity. This method can evaluate the poses of various animals, offering a robust solution to distinguish different poses of the same livestock in images.

Livestock pose estimation effectively differentiates between various poses of livestock in images, enabling the selection of appropriate poses for subsequent livestock dimension measurement. This would enhance the accuracy of measurements in livestock body dimension assessment.

In summary, prior to dimension measurement, processing livestock images usually involves several steps. Employing image processing techniques to identify the livestock species in the images, precisely capturing the livestock’s outlines, segmenting the target livestock from the background, and assessing posture to determine whether the subject conforms to the standard measurement pose are common steps. Finally, the dimensions of the body are measured. As image data usually have smaller file sizes and limited information and occupy less space, researchers can process images in a short time. However, with this approach, the measurement of certain parts of the livestock is often constrained.

### 3.2. Processing of Livestock Point Cloud Sensing Data

Point cloud data typically have larger file sizes and contain more information. Currently, the processing of livestock point cloud data is a key focus in the research of automated livestock dimension measurement. The processing of livestock point cloud data generally includes several steps, such as point cloud registration and reconstruction, point cloud segmentation, point cloud simplification, point cloud completion, point cloud posture judgment, and normalization [31].

#### 3.2.1. Point Cloud Registration and 3D Reconstruction

Three-dimensional point cloud registration and reconstruction of the livestock are generally the initial steps in livestock data processing. After obtaining the livestock’s point cloud data, the corresponding point cloud data are registered, and a three-dimensional point cloud is reconstructed to extract information for subsequent point cloud processing. Point cloud registration techniques include the ICP (Iterative Closest Point) algorithm, the NDT (Normal Distribution Transform) algorithm, and the PFH (Point Feature Histogram) algorithm. Typically, data captured from multiple viewpoints are matched to ultimately reconstruct the three-dimensional model of the livestock, as illustrated in Figure 7.

Currently, numerous high-precision livestock point cloud registration algorithms have been developed. Shi et al. (2020) [32] used a Kinect depth camera to capture point cloud data on walking pigs from three different viewpoints. The registration parameters were obtained from a bounding box, and three local point clouds were used for reconstruction. The experimental results showed that the average error in measuring the dimensions of 40 pigs was less than 4.67%. However, due to the data being acquired from freely moving pigs, which introduces noise and complex poses, the measurement algorithm requires the animals to assume standard poses. Furthermore, the limitations of the environment led to the design of barriers in the data collection setup to restrict the pigs’ movement, which might have hindered data collection. Additionally, the experimental data only covered a single pig breed, potentially limiting the system’s generalizability. Moreover, the system’s operation is not very user-friendly for ordinary workers, as it requires specialized personnel for installation and maintenance, posing practical operational challenges that may hinder meeting commercial demands. Dang et al. (2022) [33] proposed a framework for reconstructing three-dimensional point clouds of cows. This method utilized two cameras to capture the cow data, enhanced the data quality using a convolutional neural network, and then improved the point cloud registration accuracy using the SLAC algorithm. This approach requires creating 3D point clouds for each part of the cow and then forming the overall 3D cow point cloud, which reduces the translation during the registration process and enhances cow point cloud reconstruction. However, this method necessitates a sufficiently large and accurate dataset of cow data, covering a wide range of poses, and the study did not mention the time required for reconstruction. Therefore, whether it is suitable for commercialization remains to be confirmed. In the context of livestock point cloud registration and reconstruction, annotating key points in the point cloud is also an essential task. Raphael et al. (2022) [34] reformulated key point extraction as a regression problem based on the distances between the key points and the rest of the point cloud, introducing a novel method for annotating key point cloud points. Li et al. [11] developed a real-time system for collecting cow point cloud data, equipped with five depth cameras in a gantry structure. The system can collect point cloud data from five different angles within 0.09 s, and the process of reconstructing the cow’s three-dimensional point cloud takes 10–15 s. The overall error ranges from 0.5% to 2%, indicating that the reconstructed cow point cloud accurately represents its true shape. This system offers the potential for automated animal growth monitoring and, with further refinement, could be applied to ordinary farms commercially.

Point cloud registration and reconstruction constitute a crucial step in three-dimensional machine vision. This process involves merging data from various point clouds defined in local coordinate systems into a global coordinate system, resulting in a complete three-dimensional livestock point cloud model dataset.

#### 3.2.2. Point Cloud Object Extraction

Three-dimensional point cloud segmentation for livestock is an essential processing step. Typically, the point cloud data obtained using depth cameras encompass not only the livestock but also the surrounding scene elements, such as the ground and fences. Point cloud segmentation is needed to isolate the livestock point cloud from the background. Currently, there are various methods for point cloud segmentation, including random sample consensus (RANSAC), Euclidean clustering, voxel-based segmentation, and deep-learning-based segmentation.

In a study by Zhang et al. (2023) [35], after acquiring three-dimensional point clouds of cattle using depth cameras, statistical outlier removal was applied to eliminate sparse points, followed by RANSAC plane segmentation to remove ground points. Finally, a threshold-based cutting method was used to eliminate background points. While this approach produced reasonable results in terms of the cattle’s three-dimensional point cloud, it exhibited several limitations. It was constrained by the use of a single depth camera capturing data from only one perspective, leading to limited and non-universal data. Additionally, the cattle model used lacked versatility. On the other hand, Li et al. (2022) [19] also utilized depth cameras for data collection in cattle farms. Their method involved a series of algorithms including spatial filtering, statistical outlier point filtering, random-sample-consensus-based shape fitting, point cloud downsampling, and density-based clustering to remove background interference, such as from fences, while preserving the integrity of the point cloud. This approach effectively extracted point clouds of target cattle from complex real-world environments, providing crucial support for subsequent operations like livestock measurement, as shown in Figure 8.

In summary, point cloud segmentation is instrumental in isolating the relevant livestock point cloud data for further operations such as body measurement from complex and cluttered environments, contributing significantly to the accuracy of livestock-related tasks.

#### 3.2.3. Point Cloud Simplification

Livestock point cloud data typically exhibit a substantial data volume, leading to inefficiencies in data processing. To address this, a common practice involves point cloud simplification to retain the livestock’s essential features while improving the data processing efficiency [36], as depicted in Figure 9.

In a study by Zhang et al. (2019) [37], they employed Principal Component Analysis (PCA) to compute local plane normal vectors, which were combined with curvature information to simplify the point cloud of the cow’s back. This method yielded an average absolute error of less than 1.17 cm in extracting the body measurement points, along with a 33.72% reduction in the point extraction time. Such techniques provide valuable technical support for automating livestock body measurement. Qin (2020) [38] introduced an octree-based K-means clustering approach to point cloud simplification. This method not only removed redundant data from pig point clouds but also effectively retained the fine details of the pig’s features. However, the experiments were conducted solely on three-dimensional point clouds of pigs, and the general applicability of this method to other livestock types was not verified.

Point cloud simplification not only reduces the data volume but also preserves crucial livestock details. This enhancement significantly boosts the efficiency of various data operations, thereby reducing the required processing time.

#### 3.2.4. Point Cloud Filling

During the process of collecting point cloud data from livestock using devices, various factors often lead to missing points in the point clouds. For instance, when collecting data from cows, safety measures such as setting up barriers are necessary. However, these barriers can obstruct and result in missing portions of point cloud data.

Chu et al. (2020) [39] proposed a frame completion method based on video. They captured depth videos of cows and synchronized a different view’s point cloud from another frame to complete the missing areas in the side-view point cloud. While effective in compensating for the missing data caused by barriers, this method relies on manually selecting frames and involves significant computational overhead due to the depth videos, which may not align with the needs of modern livestock informatics.

Li et al. (2021) [40] presented a method for completing missing sections in side-view point clouds of pigs. This approach utilized threshold analysis, cubic spline curve fitting, and edge detection, as illustrated in Figure 10. Although it successfully addressed pig point cloud completion, the experiments were performed on model pigs, limiting its general applicability.

Yang et al. (2022) [10] initially sliced cow point clouds and hole boundary points along the principal direction of the cow’s body. They then employed a smoothing-factor-equipped spline curve to fit and fill these sliced points in polar coordinates. This completion method, compared to grid-based methods, generated points that are closer to reality, reducing the measurement errors caused by point cloud gaps in livestock data.

In summary, point cloud completion effectively resolves incomplete livestock point cloud data due to environmental factors. It enhances the adaptability of equipment during body measurement procedures, laying a robust foundation for commercialization.

#### 3.2.5. Posture Judgment and Normalization

During the process of livestock body measurement, variations in the posture of the animals can lead to different measurement results. To ensure accurate measurement of livestock dimensions, it is common to perform an assessment of an animal’s posture first. Hu et al. (2022) [41] proposed a curve skeleton extraction method specifically designed for incomplete livestock point clouds. This method extracts the skeleton, which is then used for posture assessment to determine whether a pig is in the correct posture for body measurements. While this method was tested on other livestock as well, it exhibited a relatively high error rate. Therefore, further improvements are needed when dealing with different species of animals.

Pose normalization involves transforming data from various coordinate systems into a unified coordinate system. This facilitates data comparison and processing and enhances the algorithm’s precision [42]. Guo et al. (2019) [43] introduced a pose normalization method for 3D livestock point clouds, leveraging the bilateral symmetry of cows or pigs to transform them into a standardized coordinate system. However, this method cannot standardize all livestock to the same reference pose. Lu et al. (2022) [44] improved upon this method by using 2D object detection to determine the orientation of livestock in 3D space. They then projected the 2D detection results into 3D to locate the livestock targets. Finally, they applied a bilateral symmetry-based pose normalization framework, as depicted in Figure 11, effectively normalizing the livestock poses. This 2D/3D fusion approach addresses the issues caused by noise and missing values in depth camera captures and provides stability and practicality. Luo et al. (2023) [7] collected livestock data during free movement. Given the diverse poses of the fitted mesh, they employed a shape statistical model to normalize different poses to a reference pose, thereby mitigating the posture-related measurement errors induced by animal movements. Although the iterative process of model fitting in this method is time-consuming, it can readily adapt to measuring non-rigidly joint-deformed objects, effectively addressing the issue of inconsistent livestock data due to pose normalization. In cattle body measurements, the overall estimation accuracy was 91.95%, while in pig body measurements, the accuracy was 87.63%. This approach provides a solution for accurate reconstruction and the measurement of livestock after pose normalization in precision livestock farming.

Posture assessment and normalization play a crucial role in addressing non-standard postures of livestock, reducing errors and achieving more accurate measurements of livestock dimensions.

In conclusion, the processing of livestock point cloud data plays a crucial role in the automation of livestock body measurements. After collecting data from livestock using suitable sensing devices, various processing operations, such as point cloud registration and reconstruction, point cloud segmentation, point cloud simplification, point cloud completion, point cloud posture judgment, and normalization, are employed. These operations resulted in more comprehensive and higher-quality livestock data, achieving a more precise delineation of the livestock’s contours and laying a solid foundation for the accuracy of subsequent livestock body measurements.

## 4. Livestock Body Measurement Sensing Data Analysis

Livestock body dimension measurement involves the collection and processing of livestock data, followed by the use of technologies such as digital image processing, deep learning, and computer vision to perform body measurements on the obtained 3D point cloud or image sensing data on livestock, as illustrated in Figure 12. This section primarily covers the standards for livestock body measurement and the methods of performing such measurements using 3D point cloud or image sensing data.

### 4.1. Livestock Body Dimension Measurement Standards

Body dimension measurement is a crucial indicator for assessing the growth and development of livestock. These measurements are typically expressed in numerical form using units such as length, angles, and areas to represent the size of various parts of the livestock, such as body height, body length, chest circumference, girth, etc. These measurements reflect the physical size of the livestock and hold significant importance for livestock management and breeding purposes [45,46]. Manual body dimension measurements often involve tools like measuring sticks and tape measures [47]. During the measurement process, the livestock are usually secured within a device made of iron to ensure the safety of the measurement personnel and to maintain the livestock in a proper posture. Measurements are then taken using the measurement tools, as shown in Figure 13.

Currently, there are a wealth of livestock resources available [48]. Figure 14 illustrates the measurement standards for pigs, cattle, and sheep, which include various indicators such as chest circumference, abdominal circumference, body height, body length, body width, girth, chest depth, chest width, and more. Table 5 provides specific locations on the body of pigs, cattle, sheep, etc., where specialized measurement instruments are used for measurement purposes [49,50,51]. The aforementioned morphometric parameters are commonly employed for phenotypic assessment in the breeding process.

### 4.2. Measurement of Livestock Body Dimensions Using Images Sensing Data

During the initial stages of researching livestock body intelligent dimension measurement techniques, many studies were primarily based on image sensing data due to limitations in the software and hardware and other factors. However, with the advancement of technology, image sensing data have become more diverse, leading to the emergence of various new techniques. Table 6 provides a summary of the recent and relatively typical research outcomes in the field of livestock body measurement conducted using image-based approaches.

#### 4.2.1. Body Dimension Measurement Based on Color Image Sensing Data

In the early stages of research on automated livestock body measurement, most researchers used regular RGB cameras to capture color image data on the animals. For instance, Shi et al. (2020) [52] employed image processing techniques to measure simple dimensions of cattle, achieving low measurement errors and high accuracy. However, the measurements were limited to only a few aspects of the cattle’s body, such as height and length. To extend the coverage of the livestock body measurements, some researchers adopted a multi-view approach to data collection [53,54].

In recent years, with the rapid development of new technologies like deep learning, some researchers have harnessed convolutional neural networks (CNNs) to enhance the measurement accuracy, showcasing the potential of these emerging techniques. Zhao et al. (2021) [55] utilized the Mask R-CNN algorithm to effectively extract the contour of cattle against complex backgrounds. They smoothed the contours and divided it into segments to extract feature regions. Using the chord length curvature method, they computed the maximum curvature point within each feature region to obtain cattle body measurement data. While this method effectively extracts the cattle’s contours, variations in the cattle’s standing posture can still impact the measurement results. Zhang et al. (2021) [56] introduced a data-mining-based approach to estimate the body size of yaks. The results showed that this method was convenient and efficient, offering an effective means to measure the body size of yaks.

The aforementioned studies primarily relied on RGB cameras and image processing techniques, with some assistance from technologies like deep learning. Nevertheless, there is still room for improvement in terms of the measurement coverage and accuracy.

#### 4.2.2. Body Dimension Measurement Based on Depth Image Sensing Data

Recently, with the continuous development of consumer-grade depth cameras, many researchers have turned their attention to depth images. For instance, Yuan et al. (2022) [57] employed a watershed algorithm to extract cattle from depth images. They then used the Hough transform to extract key points from the image. By utilizing polynomial curve fitting methods and leveraging the skeletal features of the cattle’s head area, they removed the head and neck regions. Lastly, based on the spatial features of the cattle’s body measurement points, they calculated various body measurement data. This method achieved a relative measurement error of less than 3.3%, effectively enhancing the accuracy of automated cattle body dimension measurements. Zhao et al. (2023) [58] proposed a non-contact method for rapid pig body measurement based on the DeepLabCut algorithm. This approach combined depth images with deep learning to calculate five body dimension measurement indicators for pigs: length, width, height, hip width, and hip height. Not only did this method increase the coverage of the body measurements but it also achieved a maximum root mean square error of 1.79 cm and a processing time of 0.27 s per frame. The approach exhibited minimal errors and even a shorter processing time.

#### 4.2.3. Body Dimension Measurement Based on the Fusion of Color and Depth Image Sensing Data

Furthermore, some researchers have combined color images with depth images. Nan et al. (2021) [59] proposed a method that uses depth and color images of cows, employing an improved template-matching approach to detect and segment body parts. This method not only achieves a high average segmentation accuracy but also offers a new avenue for cattle body measurement. Zhao et al. (2022) [60] utilized the Kinect v4 depth camera to capture color and depth images of cows. They employed deep learning YOLOv5 object detection, Canny edge detection, and three-point circular arc curvature algorithms to extract cows’ body feature points and then calculated the body measurement data. The average relative measurement error of this method does not exceed 2.14%, exhibiting high accuracy in complex real-world environments and providing a novel research direction for machine-vision-based cattle body dimension measurement.

**Table 6 sensors-24-01504-t006:** Research on image-based measurement of livestock body dimensions.

Acquisition Method	Collecting Device	Collecting Data	Object	Position	Technical Method	Research Results	Time	Literature
SP	2RC	2D image	Cows	BL, BH	Image processing	The relative errors are 2.28% and 0.06%.	2020	Shi [52]
	2RC	2D image	Cows	BL, BH, BW	Image processing	The average error is less than 1.21%.	2020	Zhang [54]
	2RC	2D image	Cows	BL, BH	Deep learning Image processing	The average relative error of on-site system validation for a certain pasture is less than 6.85%.	2020	Li [61]
	2RC	2D image Depth image	Pigs	BL, BH, etc.	Image processing	The average relative error within the normal bending range of a pig’s body is less than 2.93%.	2021	Xu [62]
	2RC	2D image	Cows	BL, CD, etc.	Image processing Data mining	The average error is within 4.91%.	2021	Zhang [56]
	DC	2D image	Cows	BL, BH	Deep learning Image processing	The average relative error is within 8.36%.	2021	Zhao [55]
	DC	2D image Depth image	Cows	BL, BH, etc.	Image processing	The average relative error is within 2.14%.	2022	Zhao [60]
	DC	Depth image	Cows	BL, BH	Deep learning Image processing	The average relative error is within 3.3%.	2022	Zhao [63]
CA	2RC	2D image	Cows	BL, BH, BW	Machine vision Image processing	The average relative error is within 3.73%.	2020	Hu [53]
SF	DC	Depth image	Cows	CW, etc.	Deep learning Image processing	The average absolute percentage error is 3.13%.	2021	Kamchen [21]
	DC	Depth image	Pigs	BL, BH, BW, etc.	Deep learning Image processing	The maximum root mean square error is 1.79 cm.	2023	Zhao [58]
	DC	Depth image	Cows	BL, BH, AG, etc.	Image processing	The average absolute error is within 2.73 cm.	2022	Ye [20]
	DC	Depth image	Cows	BL, AG, etc.	Image processing	The average relative error is within 3.3%.	2022	Chu [57]

The aforementioned developments represent recent advancements in image-based livestock body measurement. While image processing algorithms currently exhibit fast computational speeds, there are still limitations in terms of their accuracy and the coverage of the body dimension measurement areas. With the ongoing development of depth imaging, it is anticipated that these limitations will be addressed in the future. In conclusion, the technology in this field is gradually maturing and holds substantial commercial potential.

### 4.3. Body Dimension Measurement of Livestock Using 3D Point Cloud Sensing Data

With the advancement of technology, laser scanners and consumer-grade depth cameras are continually evolving and improving. Three-dimensional information acquisition techniques based on computer vision sensing theory are gradually becoming the mainstream in this field [64], Many researchers have applied three-dimensional point cloud technology to the field of livestock husbandry for measuring livestock body dimensions. Table 7 provides a summary of recent and noteworthy research outcomes in the realm of livestock body measurement based on three-dimensional point clouds.

#### 4.3.1. Body Dimension Measurement of Livestock Using Geometric Segmentation Algorithms

In the initial stages of using three-dimensional point clouds for livestock body measurement, many approaches were developed based on point cloud segmentation algorithms. Ma et al. (2020) [65] proposed an improved region growing method for segmenting sheep point clouds and then performing body measurement. This method achieved a maximum relative error of 2.36%, indicating a high measurement accuracy. However, the method relies on manually selecting measurement points during the measurement process, which introduces issues such as inaccurate point selection, slow measurement times, and cumbersome operations. Zhang et al. (2023) [35] also utilized point cloud segmentation to obtain better cattle body outlines for subsequent body dimension measurement. While this method is straightforward, its limitation lies in the fact that only one side of the cattle body is captured in the acquired point cloud data. Li et al. (2022) [66] developed a fully automated method for measuring beef cattle’s body dimensions. This approach not only achieved a high accuracy and speed in measurement but also covered a comprehensive set of measurement locations, making it a practical solution for cattle body measurement with significant commercial potential. However, when measuring the same livestock in different poses, noticeable errors were observed in the results.

#### 4.3.2. Body Dimension Measurement of Livestock Using Deep Learning Segmentation Algorithms

In recent years, with the rapid development of deep learning, there have been significant advancements in the deep learning techniques applied to three-dimensional point clouds [67]. Huang et al. (2019) [68] proposed a method for measuring the body dimensions of live cattle using deep learning. This method achieved a high accuracy with its errors within 2.36%, which generally meets the practical requirements. However, it is only suitable for cattle breeds with short to medium hair; for other cattle breeds, the measurement errors are significantly higher, making it unsuitable for commercial applications. Hu et al. (2023) [69] improved the PointNet++ model and divided entire pig point clouds into various parts, such as the head, ears, torso, limbs, and tail. They then used the point clouds from the different parts to locate and calculate key measurement points, effectively eliminating the interference caused by the other parts of the point cloud. This approach effectively addresses the issue of interference between different body parts due to pigs’ movement or other behaviors, which can affect the accuracy of measurements in areas such as height, chest circumference, and hip circumference. This method provides a more accurate measurement of pigs’ body dimensions in different poses and offers a promising approach to incorporating deep learning with three-dimensional point clouds into the field of livestock farming.

#### 4.3.3. Body Dimension Measurement of Livestock Using 2D and 3D Fusion Methods

In the present context, due to the intricate rearing environments of livestock farms, directly acquiring three-dimensional point clouds of livestock is not straightforward. Some researchers opt to initially capture two-dimensional images using regular RGB cameras and then reconstruct these multiple images into three-dimensional point clouds. For instance, Shi et al. (2022) [22] employ unmanned aerial vehicles to capture images of cattle bodies, followed by three-dimensional reconstruction. Subsequently, based on the reconstructed cattle point clouds, they performed body dimension measurements. This method primarily suits large-scale and low-density farming environments. Simultaneously, certain researchers consider utilizing images as an auxiliary during the process of three-dimensional point-cloud-based body dimension measurements. Du et al. (2022) [70] propose a method that combines two-dimensional images with three-dimensional point clouds for livestock body dimension measurements. This approach not only enables measurements of more anatomical parts but also yields more accurate results, extending its applicability across a broader range.

Currently, although three-dimensional point cloud technology is rapidly advancing and being integrated into livestock body dimension measurements, there is ongoing refinement. Numerous distinguished researchers have made substantial contributions to its development. However, challenges remain in accurately measuring livestock body dimensions across different postures. Yin et al. (2022) [71] introduce a pig posture classification algorithm and use regression models to adjust the measurements in non-standard postures, thereby enhancing the measurement precision. Li et al. (2023) [72] devise a Pose Measurement Adjustment (PMA) model for calibrating various poses of cattle, predicting the calculation errors under different postures, and significantly reducing measurement result discrepancies.

**Table 7 sensors-24-01504-t007:** Research on measurement of livestock body dimensions based on 3D point cloud.

Acquisition Method	Collecting Device	Collecting Data	Object	Position	Technical Method	Research Results	Time	Literature
SP	DC	3D point cloud	Cows	BL, BH, AG, etc.	3D Visual Technology	The maximum error is 9.36%, and the minimum error is 1.10%	2023	Zhang [35]
	DC	3D point cloud	Sheep	BL, BH, CD, etc.	3D Visual Technology	The maximum relative error is 2.36%	2020	Ma [65]
	2RC	2D image	Cows	BL, BH, CG	Image Processing 3D Visual Technology	The average relative errors are 3.87%, 4.16%, and 5.06%, respectively	2022	Shi [22]
	2RC	2D image	Cows	BL, CG, CW, etc.	Image Processing 3D Visual Technology	The average relative error is less than 4.67%	2022	Yang [10]
	2RC	2D image	Cows	BL, BH, CG	Image Processing 3D Visual Technology	The average errors are 3.34%, 3.74%, and 4.73%, respectively	2023	Chen [73]
CA	DC	Digital image 3D point cloud	Pigs Cows	BL, BH, BW, CW, etc.	Image Processing 3D Visual Technology	Reduce the average absolute percentage error to below 10%	2022	Du [70]
	DC	3D point cloud	Pigs	BL, BH, BW, AG, etc.	3D Visual Technology	The average relative error is less than 4.67%	2020	Shi [32]
	DC	3D point cloud	Pigs	BL, BH, BW, AG, etc.	Deep Learning 3D Visual Technology	The average relative error is less than 5.26%	2023	Hu [69]
	DC	3D point cloud	Cows	BL, BH, BW, CG, etc.	3D Visual Technology	The average relative error is less than 2.8%	2022	Li [66]
	DC	3D point cloud	Pigs, cows	BL, CD, CW, etc.	3D Visual Technology	For cattle body measurement, the overall estimated accuracy is 91.95%, while in pig body measurement, the accuracy is 87.63%	2023	Luo [7]

In summary, compared to measurement based on image sensing data, livestock body dimension measurements based on three-dimensional point cloud sensing data generally yield superior results and can encompass more body parts. However, due to the substantial data volume of point clouds and lengthy computation times, real-time commercial collection demands high-performance hardware. With researchers continually innovating and enhancing the algorithms, the rapid development of three-dimensional point cloud-based body dimension measurement technology is underway, gradually addressing challenges related to the processing times, costs, and commercial viability.

## 5. Discussion

### 5.1. The Current Challenges

Currently, despite significant innovations and continuous improvements by researchers in the field of the measurement of livestock body dimensions both domestically and internationally, yielding a substantial body of research outcomes, there still exist some key challenges:

In the livestock data collection process, the sensing equipment costs are generally high, restricting widespread adoption and application of the technology. The acquired point cloud data have a large volume, leading to long algorithm computation times and a low processing efficiency, which is unfavorable for automation. Moreover, the current technologies often have specific requirements for measurement scenes, limiting their applicability to various real-world scenarios.

During the measurement process, considerations for sensing equipment safety are typically necessary, leading to the addition of barriers and other protective measures around the measurement devices and personnel. However, determining the optimal placement to minimize the impact on the data collection remains a critical challenge. Additionally, the current technologies often involve measuring livestock individually, hindering scalability and efficiency in mass production. Furthermore, the accuracy of measurements in areas such as chest girth and abdominal girth is often suboptimal. Presently, most of the research has not achieved full automation and still requires manual assistance.

### 5.2. Outlook for the Future

Based on the challenges and issues identified in automated livestock body dimension measurement technology, the future key areas of research and development trends are as follows:

In the process of point cloud acquisition, further efforts should be directed at reducing costs. This can be achieved by optimizing body dimension measurement algorithms to decrease sensing device expenses, thereby facilitating better commercialization. Additionally, depth cameras can directly reduce the amount of point cloud data, simplify the point clouds, and retain the inherent features of the livestock data. Given the rapid development of the livestock industry, various ranch environments possess unique characteristics. It is hoped that future research methods can adapt to diverse livestock settings.

During the process of livestock body dimension measurement, the future research can involve collaborating with experts in livestock husbandry. By considering factors such as livestock breeds and behaviors, specialized measurement devices can be developed. These devices would safeguard equipment and personnel while minimizing any impact on the livestock. Efforts can also focus on investigating methods for batch measurement of livestock body dimensions. This approach would enhance scalability and efficiency, thus contributing to increased overall effectiveness. Furthermore, researchers can explore the integration of both livestock images and point cloud sensing data. By harnessing the complementary strengths of these data sources, the accuracy of livestock body dimension measurements can be improved. In the current landscape of advancing AI technology, the incorporation of AI-powered robots into the process of livestock body dimension measurement can be considered. This integration aims to enhance overall livestock body dimension measurement, ultimately advancing the commercialization of this technology.

## 6. Conclusions

This article primarily introduces the livestock body dimension measurement technology from three aspects: livestock sensing data acquisition, livestock sensing data processing, and livestock sensing data analysis. A comparison is made among three types of depth cameras—structured-light, binocular vision, and time-of-flight—and conventional RGB cameras regarding their advantages, disadvantages, and suitable scenarios for livestock data acquisition. Currently, livestock data collection employs three main methods: channel archway style, suspended fixed style and simple portable style. Subsequently, the handling of livestock data for both image and point cloud sensing data is discussed. A comparative analysis reveals that both sensing data acquisition and processing significantly influence the results of livestock body dimension analysis measurements. Finally, we summarize the research on livestock body dimension measurements in both the image and 3D point cloud domains, while also contrasting and analyzing the current achievements in this field. The challenges faced in current livestock body dimension sensing technology, such as low efficiency and high costs, are highlighted. Furthermore, potential future development trends in this field are proposed.

## Figures and Tables

**Figure 1 sensors-24-01504-f001:**
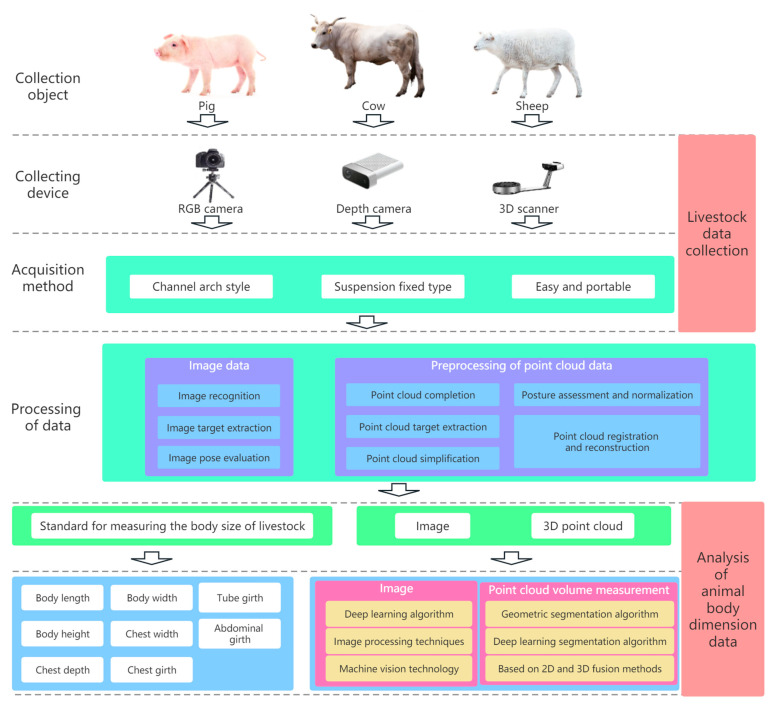
Schematic diagram of the basic principles of measuring livestock dimensions based on computer vision sensing.

**Figure 2 sensors-24-01504-f002:**
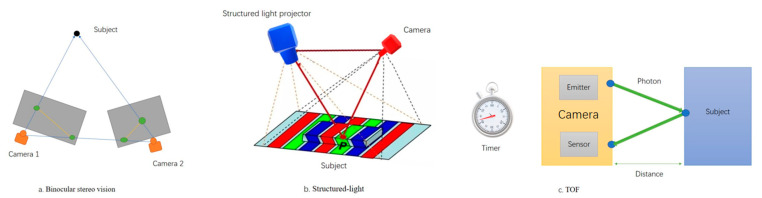
Basic principle diagram of depth camera.

**Figure 3 sensors-24-01504-f003:**
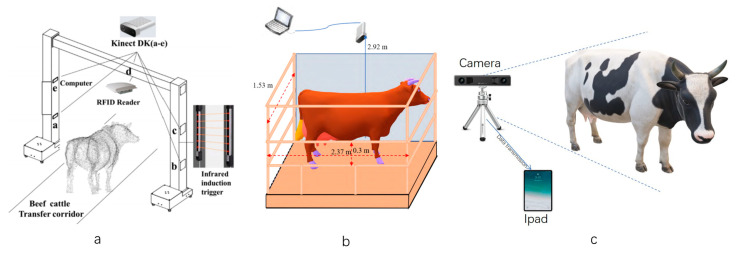
Main collection methods and equipment of livestock dimension measurement ((**a**). Channel arch type [19], (**b**). Suspension fixed type [20], (**c**). Simple and portable).

**Figure 4 sensors-24-01504-f004:**
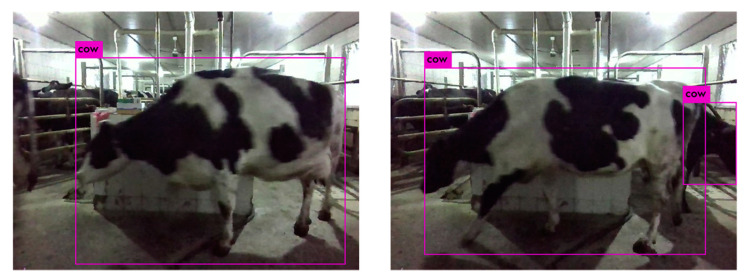
Schematic diagram of livestock image recognition [25].

**Figure 5 sensors-24-01504-f005:**
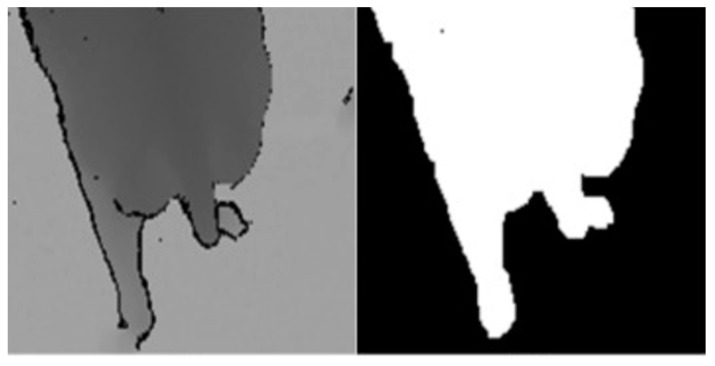
Schematic diagram of livestock target segmentation [28].

**Figure 6 sensors-24-01504-f006:**
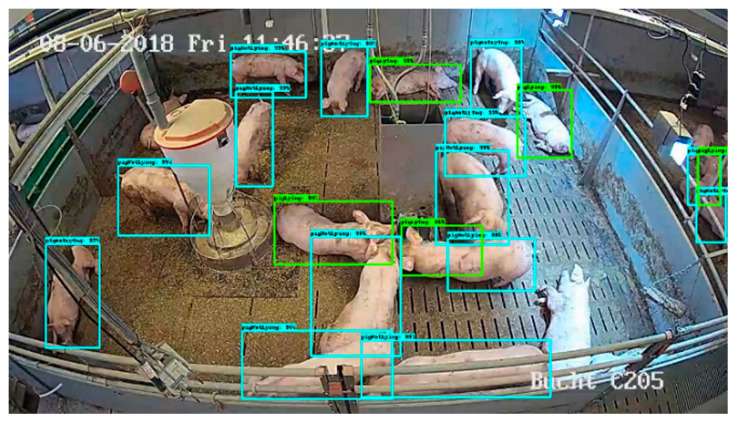
Schematic diagram of livestock posture evaluation [29].

**Figure 7 sensors-24-01504-f007:**
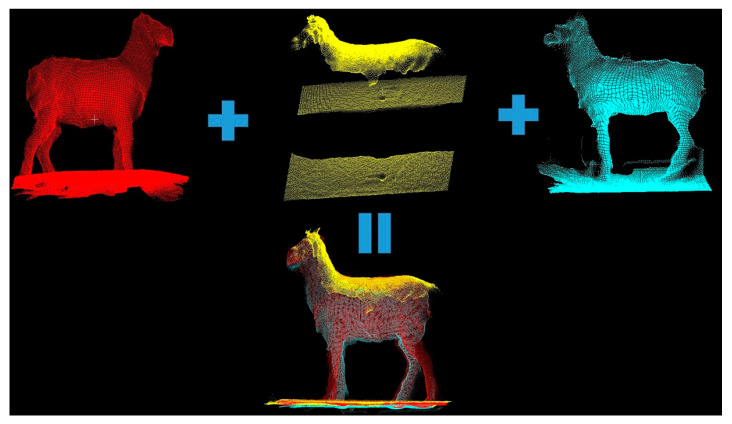
Schematic diagram of livestock cloud registration and reconstruction.

**Figure 8 sensors-24-01504-f008:**
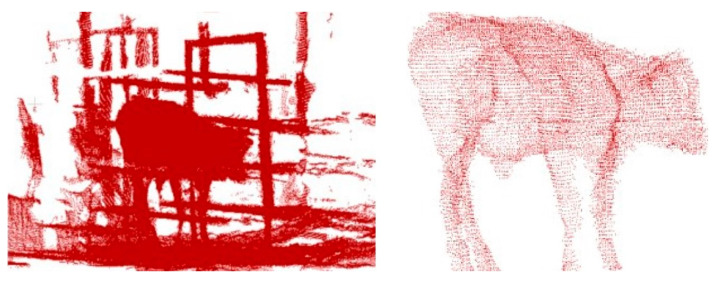
Schematic diagram of livestock target segmentation [19].

**Figure 9 sensors-24-01504-f009:**
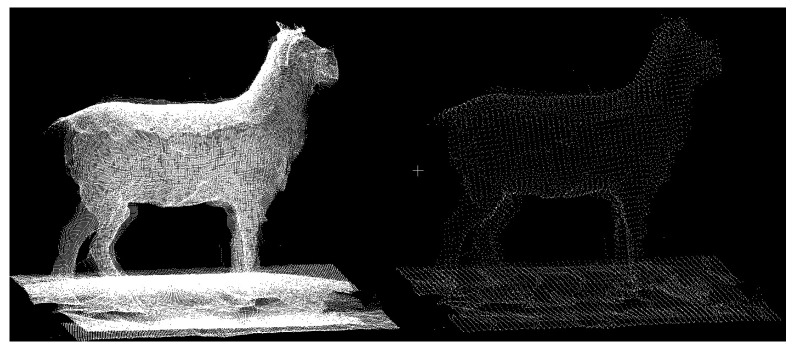
Schematic diagram of downsampling of point cloud.

**Figure 10 sensors-24-01504-f010:**
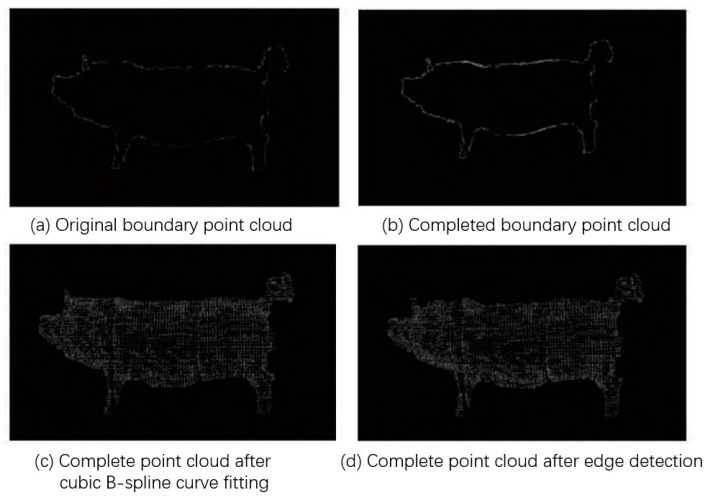
Schematic diagram of livestock point cloud completion [40].

**Figure 11 sensors-24-01504-f011:**
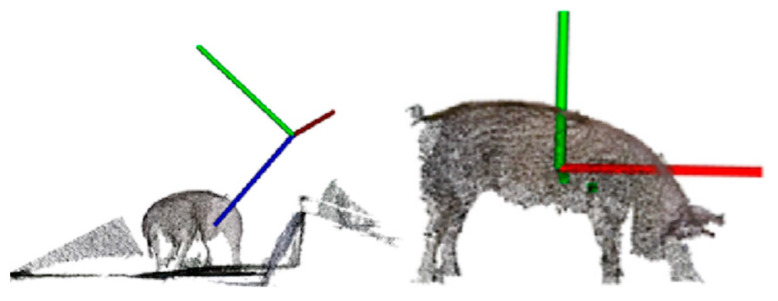
Normalization diagram of livestock point cloud posture [44].

**Figure 12 sensors-24-01504-f012:**
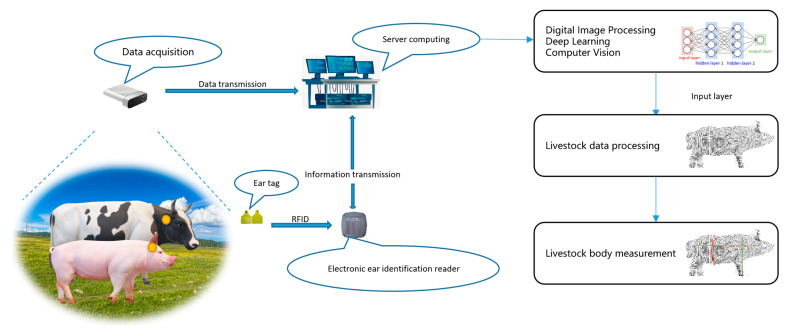
Schematic diagram of intelligent measurement of livestock body dimension.

**Figure 13 sensors-24-01504-f013:**
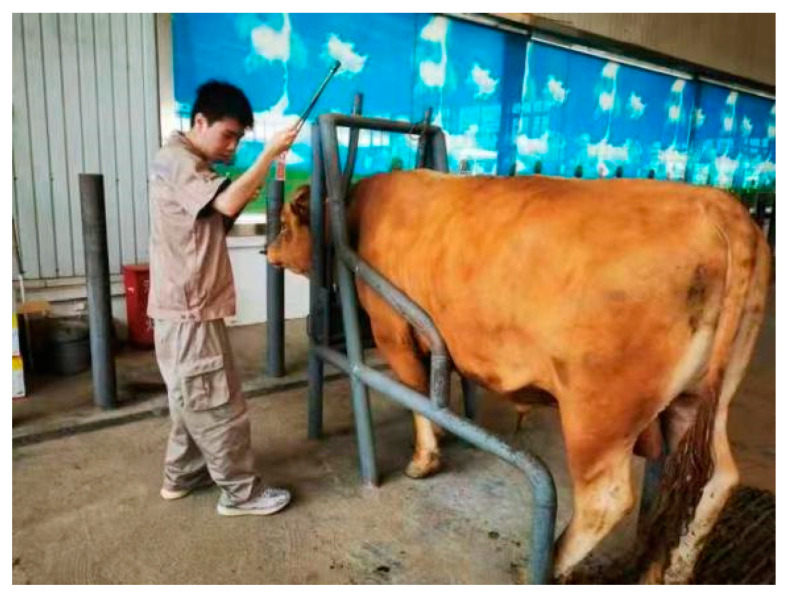
Manual measurement of livestock body dimensions.

**Figure 14 sensors-24-01504-f014:**
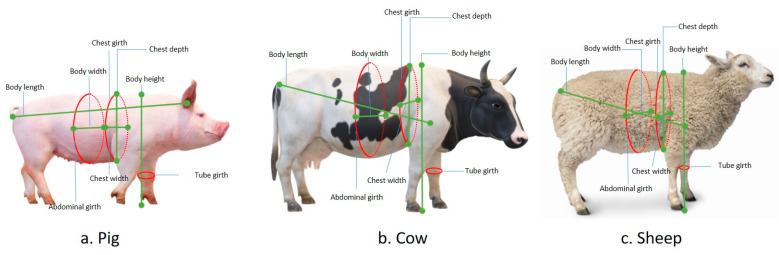
Measuring position for livestock body dimension.

**Table 1 sensors-24-01504-t001:** Comparison of sensing data collection equipment.

	Principle of Operation	Purpose	Advantage	Limitations
Depth camera (DC)	Distance from each point in the image to the camera, coupled with the two-dimensional coordinates of that point within the 2D image, derivation of the three-dimensional spatial coordinates of each point within the image	Capturing the depth distance within the specific space and spatial coordinate information	Swift processing times, spatial coordinates	Exhibits lower relative accuracy and generates larger datasets
3D scan (3S)	Scanning the spatial exterior, structure, and colors of an object, spatial coordinates of the object’s surface	Generating a high-precision point cloud representation of the object’s geometric surface	Highly accurate spatial coordinates	Long scanning process, demanding specific environmental conditions, large datasets
2D RGB camera (2RC)	An apparatus that employs the principles of optical imaging to create images	Utilizes electronic sensors to convert optical images into electronic data	Quick processing times, smaller datasets	Susceptible to environmental changes such as lighting and color

**Table 2 sensors-24-01504-t002:** Depth camera type comparison.

Depth Camera	Fundamentals	Advantages	Disadvantages	Company
Binocular stereo vision	RGB image feature point matching and indirect calculation through triangulation	Low hardware requirements, low cost, applicable indoors and outdoors, as long as lighting conditions are suitable and not too dim.	High sensitivity to ambient light, unsuitable for monotonous and texture-lacking scenes, high computational complexity, and measurement range limited by the baseline.	Leap Motion ZED DJI
Structured-light	Active projection of known encoded patterns to enhance feature-matching Effectiveness	Convenient for miniaturization, low resource consumption, active light source, usable at night, high precision within a certain range, and high resolution.	Prone to interference from ambient light, with accuracy decreasing as detection distance increases.	Apple Microsoft Intel
TOF	Direct measurement based on time-of-flight of light	Detects distant objects, with relatively minimal interference from ambient light.	High equipment demands, substantial resource consumption, low edge accuracy, constrained by resource consumption and filtering, unable to achieve high frame rates and resolutions.	Microsoft PMD Lenovo

**Table 3 sensors-24-01504-t003:** Comparison of three types of depth cameras.

	Binocular Stereo Vision	Structured-Light	TOF
Resolution	Medium–high	Middle	Low
Precision	Medium	Medium–high	Medium
Frame rate	Low	Medium	High
Anti-light (principle angle)	High	Low	Medium
Hardware cost	High	Low	Medium

**Table 4 sensors-24-01504-t004:** Main collection methods of livestock data.

Collection Method	Merit	Shortcoming
Channel Archway Style (CA)	Enables the collection of data during livestock movement, reducing stress on the animals.	Data loss due to obstruction by railings.
Suspended Fixed Style (SF)	Requires only a single camera for suspended installation, resulting in lower costs.	Single perspective of obtained livestock data, with data collection requiring the livestock to be in a stationary state.
Simple Portable Style (SP)	Convenient for transportation and easy to install.	Requires multi-angle

**Table 5 sensors-24-01504-t005:** Position definition for measuring the body dimensions of livestock.

Species	Pig	Cow	Sheep
Chest girth (CG)	The diameter of the chest is measured at the posterior corner of the shoulder blade.	Surrounds vertically around the circumference of the base of the chest.	The diameter of the chest circumference at the posterior corner of the shoulder blade.
Abdominal girth (AG)	The circumference of the largest part of the abdomen.	The circumference of the widest part of the abdomen.	The circumference of the abdomen.
Body length (BL)	The distance from the occipital ridge to the caudal root.	That is, the oblique length of the body, the straight length from the anterior edge of the shoulder end to the outer edge of the ischial end.	That is, the oblique length of the body, the straight-line distance from the anterior edge of the shoulder end to the posterior edge of the ischial tubercle.
Body height (BH)	The vertical distance from the manor to the ground.	The middle of the mane is perpendicular to the height of the ground along the posterior edge of the forelimb.	The vertical distance from the highest point of the mane to the ground.
Body width (BW)	The distance between the hips.	The horizontal maximum width of the outer edges of both hips.	The maximum horizontal distance between the hips and thighs.
Tube girth (TG)	The circumference of the thinnest part of the tubular bone.	The circumference of the upper 1/3 of the tibia of the left forelimb.	The circumference of the thinnest part of 1/3 of the tube bone.
Chest depth (CD)	The vertical distance from the mane to the lower edge of the ribs.	The shortest distance from the posterior edge of the mane to the base of the chest perpendicular.	The straight-line distance from the highest point of the nail to the lower edge of the sternum.
Chest width (CW)	The maximum distance between the vertical tangents on the left and right sides of the posterior corner of the scapula.	The minimum width behind the shoulders is measured at the same depth as the chest depth.	The straight-line distance at the widest point of the posterior edge of the shoulder blades on both sides.

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
