# Peer review of "Computer-Vision-Based Sensing Technologies for Livestock Body Dimension Measurement: A Survey"

_sensors, 2024, doi:10.3390/s24051504_

Round 1

Reviewer 1 Report

Comments and Suggestions for Authors

This paper gives a comprehensive review for the current situation of livestock body dimension measurement, from different sensors to different data types and data analysis algorithms. I think this paper will be a great help for researchers in the corresponding field. There are some small problems in the writing:

1. Line 276, the subtitle maybe "3.2.1 Point Cloud Registration and 3D Reconstruction";

2. Line 354, the subsection "3.2.3 Point Cloud Simplification" is not very relavant to the topic, as few researchers have pay attention on it now. Maybe it should be moved to the discussion part as a future challenge;

3. Line 375, "Point Cloud Completion" maybe replaced with "Point Cloud Filling".

Anyway, it is a pretty good paper. I suggest we accept it after minor revising. 

However, I still give answers to them as followings:
This paper gives a comprehensive review for the current situation of livestock body dimension measurement, from different sensors to different data types and data analysis algorithms.

This paper gives a comprehensive review for the current situation of livestock body dimension measurement, from different sensors to different data type and data analysis algorithms, which I think are original and  helpful for researchers to understand the current hot topics and future trends quickly.

It can help the researchers to set up a comprehensive view of the relevant research area and find a promising research topic.

It’s a pretty good review paper. I don’t think methodology improvements need to be added to a review paper. Only some small writing problems need to be improved.

The conclusions are good.

Comments on the Quality of English Language

The English writing is Ok.

Author Response

Dear Reviewer:
Thanks for your suggestion for this manuscript titled “Computer Vision-Based Sensing Technologies for Livestock Body Dimension Measurement: A Survey” .We have revised it according to the suggestions . If you have any other concerns regarding this manuscript please let us know as soon as possible. Thanks again for your advice and suggestions. The revised parts are marked yellow . 
Revision notes, point-to-point, are given as follows:

Comment 1:
1. Line 276, the subtitle maybe "3.2.1 Point Cloud Registration and 3D Reconstruction";
Response: Thank you for the criteria suggested. We have revised the subtitle based on your suggestion.
Modified: Line 276: 3.2.1. Point Cloud Registration and 3D Reconstruction.

We appreciate your opinions and comments.

Comment 2:
2. Line 354, the subsection "3.2.3 Point Cloud Simplification" is not very relavant to the topic, as few researchers have pay attention on it now. Maybe it should be moved to the discussion part as a future challenge;
Response: Thank you for the suggestion . We have made modifications and added the "Outlook for the Future" part as a future outlook. We think usually after the extraction of livestock point cloud targets , due to the large number of point clouds, the processing time is relatively long. Therefore, the "Point Cloud Simplification" method is necessary, which preserves the original contour features of the target point cloud and performs some point cloud simplification to facilitate subsequent data processing. Therefore, it is stated in the section “3.2 Processing of Livestock Point Cloud Sensing Data” as a bridging role and we chose to retain it.
Modified: Line 650: Additionally, depth cameras can directly reduce the amount of point cloud data, simplify point clouds, and retain the inherent features of livestock data.

We hope that the added description can be approved by you. Thanks again for your suggestions.

Comment 3:
3.Line 375, "Point Cloud Completion" maybe replaced with "Point Cloud Filling".
Response: Thank you for your valuable suggestions. We have modified the subtitle.
Modified: Line 375: 3.2.4. Point Cloud Filling

We sincerely look forward to receiving your feedback on our revisions and hope that these adjustments meet your expectations. Once again, we appreciate your patience and guidance! We tried our best to improve the manuscript and made some changes in the manuscript. 
In all, we found the reviewer’s comments quite helpful, and I revised my paper point-by-point. We appreciate for Editors and Reviewers’ warm work earnestly. Once again, thank you very much for your contribution to this manuscript.
Yours sincerely.
Dr. Ma
21 February 2024

Reviewer 2 Report

Comments and Suggestions for Authors

The work with the title "Computer Vision-Based Sensing Technologies for Livestock 2 Body Dimension Measurement: A Survey" presents a modern technology for measuring the size of animal bodies, a particularly useful method for the livestock industry. 

The work is clearly structured and includes the stages of data acquisition and processing, using three different collection methods, which are presented in the paper. Data processing is clearly presented, including: image detection, segmentation, as well as animal position detection.
The paper describes the stages of taking over and processing the 3D point clouds, data extraction, describing the factor that led to the lack of point clouds, such as the installation of barriers necessary for safety measures.  The work also presents an analysis of the data for detection regarding the measurement of animal bodies, the measurement standards, the errors obtained.  The paper ends, naturally, with discussions on current and future challenges and conclusions. In conclusion, I propose the work for publication in its current form.

Author Response

Dear Reviewer:
Thanks for your suggestion for this manuscript titled “Computer Vision-Based Sensing Technologies for Livestock Body Dimension Measurement: A Survey” .We have responded to the inquiries accordingly. If you have any other concerns regarding this manuscript please let us know as soon as possible. Thank you again for your inquiry and suggestion.

Comment 1: Could monitoring cattle traits with an unmanned aerial vehicle (UAV) help farmers monitor the condition and health of their herd without actually visiting them?
The work is interesting because it uses a Huawei P20 smartphone to capture images using SfM technology, a technology that has a lower price compared to Lidar technology. However, this method requires taking imaging data for each individual animal, increasing the animals' stress. In addition to the three collection methods presented in the paper, how do you think an unmanned aerial vehicle (UAV) could be involved for monitoring cattle traits with RGB stereo images and Lidar data from the UAV?
Response: Thank you for your inquiry. We believe it is possible, for example, Shi et al.(2022)[22] proposed a non-contact measurement method based on unmanned aerial vehicle offline 3D reconstruction to obtain cow body point cloud data. However, to ensure the quality of point cloud reconstruction, the overlap rate between the adjacent two photos should be at least 70% during the shooting process, and the reconstruction time should be long. At the same time, it is also necessary to avoid occlusion of the measurement object; Using LiDAR, although the reconstruction time is relatively short, it also faces the risk of the subject being obstructed. Therefore, currently it may only apply to low-density cattle herds raised in free range.
[22]    WEI S, YU-ZHOU C, WAN-KAI Z, et al. Cattle Point Cloud Reconstruction and Body Size Measurement System Development Based on Unmanned Air Vehicle(UAV)Platform [J]. Animal Husbandry and Feed Science, 2022, 43(04): 93-103.

Comment 2: In addition, some researchers have applied spatiotemporal interpolation techniques to remove motion noises from depth images and then detect standing animals with undefined depth values around them. Has this been tried?
Response:Thank you very much for your inquiry. But unfortunately, we haven't tried this before.It is very interesting.

Comment 3: Do you think the 3D measurement session for non-contact measurement of body conditions has a good effect and would effectively reduce the possibility of injury to animals and inspectors? Which of the presented methods do you consider relevant and how could you improve the need for data collection. What other methods could be implemented?
Response: Thank you for bringing up these questions. We believe that non-contact 3D measurement of physical condition has good results, which can effectively reduce stimulation to animals and injuries to inspectors. We believe that methods such as point cloud filtering, registration, segmentation, downsampling, 3D reconstruction, and object recognition are relevant. We can design proprietary data collection equipment based on livestock habits, collection environments, and their unique phenotypic characteristics.

Comment 4: How can the data collection method (row 149-155) be improved if more animals pass through the archway? How useful was the fixed-style method of data collection (line 156-158)?
Response: Thank you for raising these questions. We think it may be possible to design a door at the entrance where an animal enters the channel. When an animal is detected entering the channel, the door will automatically close, and when no animal is detected in the channel, the door will automatically open. Compared to arch-style data collection methods, fixed data collection methods have lower practicality and more stringent requirements. Perhaps it can be used for animal information that needs to be collected under static conditions.

Comment 5: The simple method (line 175-178) is limited, as presented in the work. What does data loss mean?
Response: Thank you for your inquiry. Data loss means that the collected animal data is incomplete and may lack data information on various parts such as the chest, legs, and head, leading to problems in subsequent body measurements or other algorithms related to animal phenotype characteristics.

Comment 6: In the process of processing the imaging data, how was the noise reduced. Were programs used? Have you tried data processing programs: Agisoft or CloudCompare?
Response: Thank you for your valuable inquiry. In the process of processing data, we attempted to use statistical filtering, direct filtering and other methods for denoising in programming. These methods can effectively remove noise points.We have also tried using CloudCompare for data processing.

Comment 7: What programs were used in this work for data processing?
Response: Thank you very much for your inquiry. In this work, we mainly use C++, based on the PCL library, to write various algorithm programs for data processing.

Comment 8: Figure 10 could be clearer, or lighter in color for a better visualization.
Response: Thank you very much for your suggestion. We have made modifications to the image.

We appreciate your patience and guidance.We appreciate for Editors and Reviewers’ warm work earnestly, and hope that the correction will meet approval. Once again, thank you very much for your comments and suggestions.
Yours sincerely.
Dr. Ma
21 February 2024